REGISTERED REPORT

# Registered report: Tumour vascularization via endothelial differentiation of glioblastoma stem-like cells

Denise Chroscinski[1], Darryl Sampey[2], Nimet Maherali[3], Reproducibility Project: Cancer Biology*†

[1]Noble Life Sciences, Gaithersburg, Maryland, United States; [2]BioFactura, Frederick, Maryland, United States; [3]Harvard Stem Cell Institute, Cambridge, Massachusetts, United States

**Abstract** The Reproducibility Project: Cancer Biology seeks to address growing concerns about reproducibility in scientific research by conducting replications of 50 papers in the field of cancer biology published between 2010 and 2012. This Registered report describes the proposed replication plan of key experiments from 'Tumour vascularization via endothelial differentiation of glioblastoma stem-like cells' by Ricci-Vitiani and colleagues, published in *Nature* in 2010 (*Ricci-Vitiani et al., 2010*). The experiments that will be replicated are those reported in Figure 4B and Supplementary Figure 10B (*Ricci-Vitiani et al., 2010*), which demonstrate that glioblastoma stem-like cells can derive into endothelial cells, and can be selectively ablated to reduce tumor progression in vivo, and Supplementary Figures S10C and S10D (*Ricci-Vitiani et al., 2010*), which demonstrate that fully differentiated glioblastoma cells cannot form functionally relevant endothelium. The Reproducibility Project: Cancer Biology is a collaboration between the Center for Open Science and Science Exchange, and the results of the replications will be published by *eLife*.

*For correspondence: joelle@scienceexchange.com

Group author details

†Reproducibility Project: Cancer Biology
See page 13

## Introduction

Glioblastoma multiforme (GBM) is a highly aggressive form of cancer characterized by an extensive network of vasculature that contributes to its invasiveness. However, the mechanisms of angiogenesis and the origin of the tumor vasculature remain poorly understood. While conventional theory suggests that GBM tumor vasculature derives from existing vessels or from bone marrow progenitor cells, recent studies have indicated that this is not the case (*Purhonen et al., 2008*). Indeed, the tumor endothelium may actually be derived from the tumor itself. GBM is maintained via a population of self-renewing, tumorigenic cancer stem cells (CSCs), which have been implicated in tumor invasion and metastasis for a large variety of cancers (*Vescovi et al., 2006*; *Fan et al., 2013*). The progeny of these CSCs are not confined to a neural lineage but rather can differentiate into functional endothelium. Based on the work of Ricci-Vitiani et al., it appears that part of the vasculature in GBM originates from tumor cells, bypassing the normal mechanisms of angiogenesis. These findings offer insights into tumor self-renewal, and offer new options for cancer treatment by targeting the differentiation of tumor cells into endothelial progeny (*Ricci-Vitiani et al., 2010*; *Kaur and Bajwa, 2014*).

In order to demonstrate the CSC lineage of GBM tumor vasculature, Ricci-Vitiani et al. first traced the genetic lineage of the tumor endothelium. They analyzed the vasculature in 15 human glioblastoma patient samples and found that a large subset of endothelial cells harbored the same mutations and chromosomal aberrations as the tumors themselves. They also showed that in culture, glioblastoma stem-like cells (GSCs) could be differentiated to express multiple endothelial markers, and showed substantial tube-forming ability, whereas fully differentiated

glioblastoma cell lines could not. Ricci-Vitiani et al. also traced the lineage of tumor vasculature in vivo, confirming the presence of human endothelial cells expressing green fluorescent protein (GFP) in a mouse xenograft of human GSCs expressing GFP. These experiments, and others, demonstrated that tumor xenografts obtained by injection of human glioblastoma neurospheres developed an intrinsic vascular network composed of tumor-derived endothelial cells (*Ricci-Vitiani et al., 2010*).

The authors next sought to investigate whether the GSC-derived endothelial cells contributed to tumor growth. This key finding is the focus of this replication study. The authors transduced glioblastoma neurospheres with a lentiviral vector containing the herpes simplex virus thymidine kinase gene (*tk*) under the control of the transcription regulatory elements of the endothelial marker *Tie2*. In this way, the tumor-derived endothelial cells could be selectively killed by exposure to ganciclovir. As negative controls, the authors used neurospheres transduced with an empty viral vector, as well as the differentiated glioblastoma cell line U87MG. As a positive control, they used neurospheres and U87MG cells transduced with a vector conferring constitutive expression of *tk* (PGK-*tk*). Upon administration of ganciclovir, selective targeting of endothelial cells generated by GSCs in mouse xenografts resulted in tumor reduction and degeneration, indicating the functional relevance of the GSC-derived endothelial vessels.

Prior to subcutaneous injection of transduced glioblastoma neurospheres, Ricci-Vitiani et al. first confirmed the lack of endogenous *Tie2* expression in both the GSCs and the U87MG cells. This quality control step will be replicated in Protocol 1, the results of which will be compared to Figure S11C. Next, the viral transduction of GSCs and U87MG cells with expression constructs for Tie2-*tk* and PGK-*tk*, as well as empty viral vector, will be replicated in Protocol 2. The generation and analysis of xenografts using various cell lines in immunocompromised mice will be replicated in Protocol 3. This protocol will generate data that can be compared to original data presented in Figures 4B, S10B, S10C, and S10D.

Recently, multiple other studies have explored the phenomenon of tumor-derived vasculature in GBM. Two recent reports found results very similar to those of Ricci-Vitiani et al., showing that oncogene-induced glioblastoma tumors gave rise to tumor-derived endothelial cells, as indicated by GFP expression. These studies also found that endothelial cells within tumors harbored the same genetic signature as the tumor itself (*Wang et al., 2010*; *Soda et al., 2011*). Similarly, Chiao et al. reported that GCSs formed vasculogenic mimicry in tumor xenografts and expressed pro-vascular molecules (*Chiao et al., 2011*). However, other groups have found that endothelial cells comprising GBM vasculature do not share the same genetic make-up as neoplastic tissues, and that GSCs routinely do not give rise to endothelial cells (*Rodriguez et al., 2012*; *Cheng et al., 2013*). Interestingly, Cheng et al. present an alternative hypothesis to that of Ricci-Vitiani et al. by showing that GSCs can give rise to vascular pericytes—which also express Tie2 (*De Palma et al., 2005*)—rather than endothelial cells. Targeting these GSC-derived pericytes disrupted vessel function and inhibited tumor size similarly as the results presented by Ricci-Vitiani et al. for targeting endothelial cells (*Cheng et al., 2013*). El Hallani et al. demonstrated that GSCs were capable of vasculogenesis in vitro, and that a fraction of GSCs could transdifferentiate into vascular smooth muscle-like cells. However, their later work suggested that rather than transdifferentiating, the GSCs were fusing with endothelial cells to create a hybrid tumor vasculature (*El Hallani et al., 2010*; *El Hallani et al., 2014*). Conversely, using a GSC mouse xenograft, Lathia et al. did not observe the integration of tumor-derived cells into the vascular wall; however, this observation was only reported in the text. No data were shown (*Lathia et al., 2011*). Ghanekar et al. tried analogous experiments using hepatocellular carcinoma CSCs and did not find any evidence that the tumor cells gave rise to the endothelium (*Ghanekar et al., 2013*).

## Materials and methods

Unless otherwise noted, all protocol information was derived from the original paper, references from the original paper, or information obtained directly from the authors.

### Protocol 1: Evaluation of *Tie2* expression in various cell lines using qPCR

This protocol evaluates the expression of the endothelial marker *Tie2* in three cell lines using semi-quantitative PCR: patient-derived glioblastoma neurospheres (GSC83), human glioblastoma cell line U87MG, and normal human dermal microvascular endothelial cells (HMVEC-d). The expression of *Tie2* will be normalized against the endogenous expression of 18S rRNA. Expression of *Tie2* is expected to be very low in GSC83 and U87MG cells, and robust in the endothelial cell line HMVEC-d, as depicted in Figure S11C. This protocol serves as a quality control step to ensure the lack of *Tie2* expression in the glioblastoma cell lines used later in the study.

## Sampling

1. This experiment will be performed three times (biological replicates) with each run using two technical replicates, for a final power of at least 80%.
   A. Test conditions:
   i. qRT-PCR of *Tie2* (and 18S rRNA) from GSC83 glioblastoma neurospheres.
   ii. qRT-PCR of *Tie2* (and 18S rRNA) from U87MG cells.
   iii. qRT-PCR of *Tie2* (and 18S rRNA) from HMVEC cells.

## Materials and reagents

| Reagent | Type | Manufacturer | Catalog # | Comments |
| --- | --- | --- | --- | --- |
| GSC83 glioblastoma neurospheres | Human cell line | N/A | N/A | Reagent being provided by original authors |
| U87MG glioblastoma cells | Human cell line | N/A | N/A | Reagent being provided by original authors |
| Normal human dermal microvascular endothelial cells (HMVEC-d) | Human cell line | N/A | N/A | Reagent being provided by original authors |
| 25 cm$^2$ tissue culture flasks | Labware | Corning | 3289 | Original brand not specified |
| Dulbecco's Modified Eagle's medium (DMEM)/F-12, no glutamine | Cell culture reagent | Sigma-Aldrich | D6421 | Replaces Gibco cat. no. 21331-020 used in original study |
| Human recombinant epidermal growth factor (EGF) | Cell culture reagent | Sigma–Aldrich | E5036 | Replaces Peprotech cat no. AF-100-15 used in original study |
| Human recombinant basic fibroblast growth factor (FGF2) | Cell culture reagent | Sigma–Aldrich | F0291 | Replaces Peprotech cat no. AF-100-18B used in original study |
| Glutamine | Cell culture reagent | Sigma–Aldrich | G7513 | Replaces Gibco cat no. 25030-081 used in original study |
| Glucose | Cell culture reagent | Sigma–Aldrich | 49163 | |
| Putrescine | Cell culture reagent | Sigma–Aldrich | P5780 | |
| Progesterone | Cell culture reagent | Sigma–Aldrich | P6149 | |
| Sodium selenite | Cell culture reagent | Sigma–Aldrich | S9133 | |
| Insulin | Cell culture reagent | Sigma–Aldrich | I9278 | |
| Transferrin | Cell culture reagent | Sigma–Aldrich | T8158 | |
| Dulbecco's Modified Eagle's medium (DMEM), high glucose | Cell culture reagent | Sigma–Aldrich | D6429 | |
| Fetal bovine serum (FBS) | Cell culture reagent | Sigma–Aldrich | F2442 | |
| Endothelial Growth Medium-2 Microvascular (EGM-2MV) | Cell culture reagent | Lonza | CC-3202 | |
| TRI reagent | Reagent | Sigma–Aldrich | T9424 | Replaces Invitrogen cat. no. 15596-018 (Trizol) used in original study |
| First-Strand cDNA Synthesis Kit | cDNA synthesis | GE Healthcare (Sigma-Aldrich) | GE27-9261-01 | Replaces Invitrogen cat. no. 28025-013 used in original study |
| 96-Well qPCR plates | qPCR | Specific brand information will be left up to the discretion of the replicating laboratory and recorded later | | |
| TaqMan Gene Expression Master Mix | qPCR | Applied Biosystems | 4369016 | |
| TaqMan Gene Expression Assay Hs00945155_m1 | TaqMan probe | Applied Biosystems | 4331182 | |
| 18S rRNA Endogenous Control 4319413E | TaqMan probe | Applied Biosystems | 4319413E | |
| StepOnePlus Real-Time PCR System | Equipment | Applied Biosystems | | |

## Procedure

Note: All cell lines will be sent for STR profiling and mycoplasma testing.

1. Culture GSC83 glioblastoma neurospheres, U87MG cells, and HMVEC cells in 25 cm$^2$ tissue culture flasks at 37°C at 5% $CO_2$.

 A. GSC83 cells should be plated at 20,000 cell/ml and subcultured once every 7 days at the same plating number. One week is sufficient time for two doublings to occur.
 i. Cells should be cultured in stem cell medium consisting of serum-free Dulbecco's Modified Eagle's Medium (DMEM)/F-12 containing:
 a. 20 ng/ml human recombinant epidermal growth factor (EGF).
 b. 10 ng/ml human recombinant basic fibroblast growth factor (FGF2).
 c. 2 mM glutamine.
 d. 0.6% glucose.
 e. 9.6 µg/ml putrescine.
 f. 6.3 ng/ml progesterone.
 g. 5.2 ng/ml sodium selenite.
 h. 0.025 mg/ml insulin.
 i. 0.1 mg/ml transferrin.
 B. U87MG cells should be cultured in 25 cm$^2$ tissue culture flasks in DMEM with 10% FBS.
 i. Subculture cells at a ratio of 1:2 to 1:5; renew medium 2–3 times per week.
 C. HMVEC-d (normal human dermal microvascular endothelial cells) should be cultured in 25 cm$^2$ tissue culture flasks in Endothelial Growth Medium-2 Microvascular (EGM-2MV).
 i. Subculture cells when they are 70–80% confluent; change growth media every other day.

2. Split each cell line into three separate 25 cm$^2$ flasks. These separate flasks constitute biological replicates for eventual downstream gene expression analysis.
 A. Allow cells to grow to log phase.

3. Isolate total RNA from the cells in each 25 cm$^2$ flask (nine flasks in total) according to the manufacturer's instructions for TRI reagent. For each sample, harvest the entire population of cells in the flask.
 A. Report total concentration and purity of isolated total RNA.

4. Reverse transcribe mRNA to cDNA according to the manufacturer's protocol.
 A. Use 500 ng total RNA for each 20 µl reaction.
 B. Use oligo(dT)$_{12–18}$ primer for first-strand synthesis.
 C. Add ribonuclease inhibitor at suggested step in the protocol.
 D. Perform RNase H digestion at suggested step in the protocol.

5. Perform qPCR to assess *Tie2* expression levels across cell types using a StepOnePlus Real-Time PCR System. Use 18S rRNA as an endogenous control. Perform duplicate technical replicates for each biological replicate (3 biological × 2 technical × 2 genes = 12 wells per cell line).
 A. Use 1 µl of undiluted cDNA mixture for each reaction.
 B. Use TaqMan probes for *Tie2* and 18S rRNA (see reagent table).
 C. Use an initial denaturation at 95°C for 10 min, following by 40 cycles of 95°C for 15 s; 60°C for 1 min.

6. Analyze and compute $\Delta\Delta C_T$ values.

## Deliverables

1. Data to be collected:
 A. Purity ($A_{260/280}$ and $A_{260/230}$ ratios) and concentration of isolated total RNA from cells.
 B. Raw qRT-PCR values, as well as analyzed $\Delta\Delta C_T$ values and bar graph of *Tie2* mRNA normalized to control mRNA levels for each condition (compare to Figure S11C).

## Confirmatory analysis plan

This replication attempt will perform the statistical analyses listed below, compute the effect sizes, compare them against the reported effect size in the original paper and use a meta-analytic approach to combine the original and replication effects, which will be presented as a Forest plot.

1. Statistical analysis of the replication data:
 A. One-way ANOVA to analyze the means of GSC83, U87MG, and HMVEC.
 i. We will then perform a Fisher's LSD test to perform multiple pairwise comparisons:
 a. GSC83 compared to HMVEC.
 b. U87MG compared to HMVEC.
 c. GSC83 compared to U87MG (sensitivity).

## Known differences from the original study

In the original study, multiple human glioblastoma neurospheres were screened for *Tie2* expression. The human glioblastoma cell lines U251 and T98G were also analyzed, as well as the human endothelial cell line HUVEC. This replication study will be using a single established glioblastoma neurosphere cell line (GSC83) provided by the authors. The authors will also provide their U87MG and HMVEC cell lines. All known differences in reagents and supplies are listed in the 'Materials and reagents' section above, with the originally used item listed in the 'Comments' section. All differences have the same capabilities as the original and are not expected to alter the experimental design.

## Provisions for quality control

The cell lines used in this experiment will undergo STR profiling to confirm their identity and will be sent for mycoplasma testing to ensure there is no contamination. The sample purity ($A_{260/280}$ and $A_{260/230}$ ratios) of the isolated RNA from each sample will be reported. All data obtained from the experiment—raw data, data analysis, control data, and quality control data—will be made publicly available, either in the published manuscript or as an open access dataset available on the Open Science Framework project page for this study (https://osf.io/mpyvx/).

## Protocol 2: Lentiviral infection of glioblastoma cells and stable cell generation

This protocol describes the methods necessary to virally transduce GSC83 glioblastoma neurospheres, as well as U87MG cells, with thymidine kinase expression constructs. The protocol first details the production of three different lentivirus strains (PGK-*tk*, Tie2-*tk*, and an empty viral vector), and then explains the techniques necessary to transduce the two human glioblastoma cell lines. Finally, the protocol includes methodology associated with assessing the transduction efficiency of glioblastoma cell lines via flow cytometry analysis as a quality control check.

## Materials and reagents

| Reagent | Type | Manufacturer | Catalog # | Comments |
|---|---|---|---|---|
| GenElute Endotoxin-free Plasmid Maxiprep Kit | Reagent | Sigma–Aldrich | PLEX15-1KT | Original brand not specified |
| pCMV-dR8.74 | Viral packaging vector | N/A | N/A | Reagent being provided by original authors |
| pMD2G | Viral packaging vector | N/A | N/A | Reagent being provided by original authors |
| pRRLsin.Tie2p.TK.PGKp.GFP.spre | DNA construct | N/A | N/A | Reagent being provided by original authors |
| pRRLsin.PGKp.TK.PGKp.GFP.spre | DNA construct | N/A | N/A | Reagent being provided by original authors |
| pRRLsin.PGKp.GFP.spre | DNA construct | N/A | N/A | Reagent being provided by original authors |
| *Bam*HI | Restriction enzyme | Sigma–Aldrich | R0260 | |
| *Nde*I | Restriction enzyme | Sigma–Aldrich | R5509 | |
| *Spe*I | Restriction enzyme | Sigma–Aldrich | R5257 | |
| *Sma*I | Restriction enzyme | Promega | R6121 | |
| 75 cm$^2$ tissue culture flask | Labware | Corning | 430641U | Original brand not specified |
| HEK293T cells | Cell line | ATCC | CRL-3216 | |
| Dulbecco's Modified Eagle's Medium (DMEM), high glucose | Cell culture reagent | Sigma–Aldrich | D6429 | Original brand not specified |
| Fetal bovine serum (FBS) | Cell culture reagent | Sigma–Aldrich | F2442 | Original brand not specified |
| GSC culture media | See reagent list from Protocol 1 for a complete list of medium components | | | |
| 2× HEPES buffered saline (HBS) | Reagent | Sigma–Aldrich | 51558 | Replaces laboratory-made buffer used in original study. pH of substituted buffer is 7.2; pH of original buffer was 7.05 |
| Calcium chloride dihydrate | Reagent | Sigma–Aldrich | C7902 | Original brand not specified |
| Hexadimethrine bromide (polybrene) | Cell culture reagent | Sigma–Aldrich | 107689 | Original brand not specified |
| 6-Well tissue culture plates | Labware | Corning | 3516 | Original brand not specified |
| 7-Amino actinomycin D (7-AAD) | Flow reagent | Life Technologies | A1310 | Original brand not specified |
| Tubes used for flow cytometry | Specific brand information will be left up to the discretion of the replicating laboratory and recorded later | | | |
| FACSCalibur flow cytometry instrument | Equipment | Becton Dickinson | | |

## Sampling

1. Outline of experimental endpoints:
   A. At the end of this protocol, we will have generated GSC83 glioblastoma neurospheres and U87MG cells stably expressing:
   i. Empty vector.
   ii. PGK-*tk* transduced.
   iii. Tie2-*tk* transduced.
   B. Total: six stable cell lines.

## Protocol

1. Grow and prepare endotoxin-free plasmid constructs according to the manufacturer's protocol for the GenElute Endotoxin-free Plasmid Maxiprep Kit.
   A. Viral packaging vectors:
   i. pCMV-dR8.74 (~50 μg DNA needed for production of three viruses).
   ii. pMD2G (~30 μg DNA needed production of three viruses).
   B. DNA construct expression vectors:
   i. pRRLsin.Tie2p.TK.PGKp.GFP.spre (~30 μg DNA needed for virus production).
   ii. pRRLsin.PGKp.TK.PGKp.GFP.spre (~30 μg DNA needed for virus production).
   iii. pRRLsin.PGKp.GFP.spre (~30 μg DNA needed for virus production).
2. Perform restriction digestions on an aliquot of purified plasmid to check vector integrity for Tie2-*tk* and PGK-*tk* plasmids. Following digestion, run digested bands on an agarose gel to visualize band pattern.
   A. For Tie2-*tk* vector (12.13 kb), digest vector with *Nde*I, *Bam*HI, and *Sma*I.
   i. *Nde*I + *Bam*HI = 1.44 kb segment (identifies Tie2 promoter).
   ii. *Bam*HI + *Sma*I = 1.62 kb segment (identifies TK insert).
   B. For PGK-*tk* vector (10.06 kb), digest vector with *Spe*I, *Bam*HI, and *Sma*I.
   i. *Spe*I + *Bam*HI = 257 bp segment (identifies PGK promoter).
   ii. *Bam*HI + *Sma*I = 1.15 kb segment (identifies TK insert).
3. On Day 1 of viral production, plate $1.2 \times 10^6$ HEK293T cells in a 75 cm$^2$ tissue culture flask.
   A. HEK293T cells should be maintained in DMEM supplemented with 10% FBS at 37°C with 5% $CO_2$.
4. On Day 2, replace the cell medium with 18 ml of fresh medium. Prepare a transfection master mix for each of the DNA construct vectors.
   A. Assemble the following components in a 15 ml polypropylene tube in the following order:
   i. 20 μg of DNA construct expression vector.
   ii. 13 μg of pCMV-dR8.74 packaging vector.
   iii. 7 μg of pMD2G envelope vector.
   iv. 150 μl of 2M $CaCl_2$.
   v. Bring volume to 1 ml with dd$H_2O$.
   vi. Add 1 ml of 2× HEPES buffered saline (HBS) and aerate solution for 20–30 s with a 2 ml pipette.
5. Immediately add the 2 ml transfection master mix directly to HEK293T cells by dropping slowly and evenly into the media, covering as much of the flask as possible.
   A. Do not mix.
   B. Place flasks in a 37°C incubator for 14–16 hr.
6. Day 3: after 14–16 hr, change media to remove DNA precipitate.
7. Day 4: 48 hr after transfection, collect viral supernatant, filter through a 0.45 μm syringe filter, and freeze in liquid nitrogen. Store at −80°C until use.
8. Culture GSC83 glioblastoma neurospheres and U87MG cells as described in Protocol 1.
   A. Plate 150,000–200,000 cells in a 6-well plate.
   B. Cells should be exponentially growing at time of lentiviral infection.
9. Infect GSC83 neurospheres and U87MG cells with lentivirus:
   A. Add viral supernatant (1 ml/50,000 cells) along with 4 μg/ml polybrene to each well.
   B. Spin plate of cells at 1800 rpm in centrifuge for 45 min.
   C. Incubate cells for 75 min in a 5% $CO_2$/37°C incubator.
   D. Wash cells twice with culture medium, then add fresh serum-free media.
   E. Seed cells into a 25 cm$^2$ tissue culture flask at 20,000 cells per ml.
   F. Allow cells to grow for 48 hr.
10. Evaluate infection efficiency at 48 hr post infection by flow cytometry.
    A. Remove an aliquot of cells (20,000–50,000 cells) from each flask. Untransduced cells (both GSC83 and U87MG) should also be prepared for use as a negative control.
    i. Pull down the cells by centrifuging each flask at ≤1000 rpm.
    ii. Remove the supernatant, leaving approximately 150–200 μl of media in the flask.
    iii. Use a P200 pipette to gently dissociate the cells. Pipette up and down several times to obtain a single cell suspension.
    iv. Save an aliquot for flow analysis, and passage the remaining cells into a new flask to expand them for further experiments.
    B. Incubate the freshly dissociated cells for 5 min with 7-amino actinomycin D (7-AAD; final concentration 5 μg/ml).
    C. Analyze cells for GFP expression using a FACSCalibur instrument. Apply the following sequential gates to the dot plots to select viable cells:

  i. FSC area/SSC area.
  ii. SSC width/SSC area to exclude aggregates.
  iii. FSC area/7-AAD area to select viable cells.
  iv. Untransduced cells serve as a negative control.
 D. Plot fluorescent protein expression in gated cells using bivariate plots.
11. Determine the percentage of transduced cells positive for GFP reporter expression in each population of cells.
 A. Exclusion criteria: Expression should be ≥80% positive in each transduced population in order for cells to be used for xenograft injection.
12. Continue to expand and culture cells until ready for injection into immunocompromised mice.

## Deliverables

1. Data to be collected:
 A. Purity ($A_{260/280}$ and $A_{260/230}$ ratios) and concentration of plasmid DNA.
 B. Agarose gel image of restriction-digested Tie2-*tk* and PGK-*tk* plasmids with molecular weight marker.
 C. FACS plots of virally-transduced GSC83 and U87MG cells.
 D. Achieved transduction efficiency (%GFP$^+$ cells) for GSC83 and U87MG cells.
2. Samples delivered for further analysis:
 A. Infected neurosphere and U87MG clones (see Protocol 3).

## Confirmatory analysis plan

1. Statistical analysis of the replication data:

 Not applicable.

## Known differences from the original study

All known differences in reagents and supplies are listed in the 'Materials and reagents' section above, with the originally used item listed in the 'Comments' section. All differences have the same capabilities as the original and are not predicted to alter experimental outcome.

## Provisions for quality control

Endotoxin-free plasmid DNA for expression constructs will be analyzed for concentration and purity. In order to verify the construction of Tie2-*tk* and PGK-*tk* constructs, restriction digestion mapping will be performed. Banding pattern will be compared to expected band sizes based on plasmid maps received from the original authors. Flow cytometry data will be analyzed using the software package FlowJo and the achieved transduction efficiency (%GFP$^+$ cells) will be calculated for each infected cell population. Untransduced, wild-type cells (both GSC83 and U87MG) will serve as a negative control for flow cytometry. 7-AAD will be used to exclude dead cells from flow analysis. All data obtained from the experiment—raw data, data analysis, control data, and quality control data—will be made publicly available, either in the published manuscript or as an open access dataset available on the Open Science Framework project page for this study (https://osf.io/mpyvx/).

## Protocol 3: Monitoring xenograft tumor size after selective targeting of cells with ganciclovir

This protocol is designed to test whether GSC-derived endothelial cells can contribute to tumor growth in vivo. Virally transduced glioblastoma neurospheres expressing the herpes simplex virus thymidine kinase gene (*tk*) under the control of the endothelial marker *Tie2* are subcutaneously injected into immunocompromised mice. Following tumor formation, mice are treated with ganciclovir, which selectively kills any cells expressing *tk*. Negative controls include neurospheres transduced with an empty viral vector, as well as the differentiated glioblastoma cell line U87MG, which should not give rise to endothelial cells. Positive controls include neurospheres and U87MG cells transduced with a vector conferring constitutive expression of *tk* (PGK-*tk*), which should target all tumor cells. Selective targeting of *tk*-expressing tumor cells should result in tumor reduction and degeneration, indicating the functional relevance of the GSC-derived endothelial vessels (as shown in Figures 4B and S10B).

## Sampling

1. These experiments will utilize at least three mice per treatment group, for a minimum power of 80%.
 A. See 'Power calculations' section for details.

   B. As per Ricci-Vitiani et al., U87MG cells have a 100% tumor incidence rate, while GSC83 neurospheres have a ~60% tumor incidence rate.
   i. To ensure that we have enough animals at the end of the study to accurately power the effects for the GSC83 mouse cohort, we are including two extra mice per group beyond the estimated sample size of our power calculations.
2. Outline of experimental conditions:
   A. Mouse Cohort 1 (xenograft of GSC83 glioblastoma neurospheres).
   i. 28 female CD1 athymic nude mice, 5 weeks old.
   a. Seven mice injected with untransduced GSC83 cells.
   b. Seven mice injected with empty vector transduced GSC83 cells.
   c. Seven mice injected with PGK-*tk* transduced GSC83 cells.
   d. Seven mice injected with Tie2-*tk* transduced GSC83 cells.
   B. Mouse Cohort 2 (xenograft of U87MG cells).
   i. 12 female CD1 athymic nude mice, 5 weeks old.
   a. Three mice injected with untransduced U87MG cells.
   b. Three mice injected with empty vector transduced U87MG cells.
   c. Three mice injected with PGK-*tk* transduced U87MG cells.
   d. Three mice injected with Tie2-*tk* transduced U87MG cells.

## Materials and reagents

| Reagent | Type | Manufacturer | Catalog # | Comments |
|---------|------|--------------|-----------|----------|
| 4–5 Week old female CD1 athymic nude mice | Mouse line | Charles River | 086 | |
| Ganciclovir | Drug | Sigma–Aldrich | G2536 | |
| Matrigel Matrix High Concentration (HC), phenol red-free | Cell culture reagent | Corning | 354262 | Original catalog number not specified |
| Dulbecco's phosphate buffered saline (PBS) | Reagent | Sigma–Aldrich | D8537 | |
| 1 ml insulin syringe with attached needle; 29G × 1/2 in. | Labware | BD Biosciences | 329411 | Original brand not specified |
| IsoFlo (isoflurane, USP) | Anesthetic | Abbott Animal Health | 05260-05 | |
| Paraformaldehyde | Reagent | Sigma–Aldrich | 158127 | Original brand not specified |
| Paraffin | Reagent | Specific brand information will be left up to the discretion of the replicating laboratory and recorded later | | |
| Xylene | Reagent | | | |
| Ethanol | Reagent | | | |
| Carazzi's hematoxylin | IHC stain | | | |
| Eosin | IHC stain | | | |
| Permount | Mounting medium | | | |

## Protocol

1. Prepare virally transduced cells for injection into mice. Perform all steps under sterile conditions.
   A. Dissociate cells in flasks to form a single-cell suspension:
   i. Pull down the cells by centrifuging each flask at ≤1000 rpm.
   ii. Remove the supernatant, leaving approximately 150–200 μl of media in the flask.
   iii. Use a P200 pipette to gently dissociate the cells. Pipette up and down several times to obtain a single cell suspension.
   B. Resuspend $1 \times 10^6$ dissociated cells in 0.1 ml cold PBS, then mix with an equal volume of cold Matrigel. Total volume should be 0.2 ml.
2. Subcutaneously inject 4–5 week old female athymic nude mice into the rear flank. Each mouse should receive a single injection.
   A. Mice should be microchipped prior to injection, so that they can be easily monitored throughout the duration of the study.
   B. For each cell type, inject the 0.2 ml cell/Matrigel mixture subcutaneously into the flanks of the mice using a 29G insulin syringe.

3. Allow tumor nodules to form. The estimated time for tumor formation for U87MG cells is 3–4 weeks. The estimated time for tumor formation for GSC83 neurospheres is 4–6 months.
   A. Check for tumors weekly and measure diameter and volume.
      i. Calculate tumor volume as (length × width$^2$)/2.
   B. Note time for tumor nodules to form as well as tumor diameter and volume.
   C. Once tumor size reaches ~10 mm in diameter, proceed to ganciclovir treatment.
4. Inject mice with 50 mg/kg of ganciclovir (GCV) per day for 5 days in total.
   A. Immediately prior to injection, record final tumor diameter and volume measurements. These data will serve as baseline controls for downstream analyses.
   B. Prepare a 5 mg/ml solution of GCV using sterile water under sterile conditions. Filter solution through a 0.2 µm sterile filter.
   C. Inject 0.2 ml of sterile solution (containing 1 mg of GCV) intraperitoneally (i.p.) into mice.
      i. This injection amount assumes an approximate mouse weight of 20 g. For mice that are larger or smaller, the injection volume should be adjusted accordingly.
5. Measure tumor size twice weekly following GCV injection for 4 weeks using calipers.
   A. Record both tumor diameter and volume. Measure tumor in two directions with calipers. Calculate tumor volume as (length × width$^2$)/2.
6. Four weeks after last GCV injection, take final tumor measurements and euthanize mice. Harvest a random subset of tumors for histological analysis.
   A. Randomly choose one mouse from each treatment group to harvest tumor tissue (eight mice in total; four mice from each cohort).
      i. Prior to euthanasia, deeply anesthetize animals using isofluorane and transcardially perfuse mice with sterile saline, followed by 4% paraformaldehyde (PFA) in PBS.
      ii. Excise tumor nodules under an operating microscope.
      iii. Fix excised tumors in 4% PFA for 24 hr at 4°C.
7. Process, embed, and mount tissues on slides.
   A. Dehydrate tissues through graded alcohols and clear in xylene.
   B. Infiltrate with, and then embed, tissues in paraffin and section into 3 µm sections.
   C. Mount sections onto positively charged glass slides.
      i. Mount a total of two sections for each tumor onto a single slide.
8. Perform H&E staining by hand using the following procedure:
   A. Deparaffinize sections twice in xylene, then rehydrate through graded alcohols (95%, 70%, 50% ETOH) to water.
   B. Stain sections with Carazzi's hematoxylin, then rinse slides in water.
   C. Stain sections with eosin.
   D. Dehydrate sections through graded alcohols (50%, 70%, 90%), and then place in xylene.
   E. Apply coverslips to slides with Permount and store slides at room temperature.
9. Blindly image stained sections and have images blindly analyzed by a board certified veterinary pathologist to verify the tumor composition of the tissue sections and analyze vascular structures for endothelial vacuolization and disruption.

## Deliverables

1. Data to be collected:
   A. Xenograft transplant records (mouse health records [age, weight], injection location, time for nodules to form, etc.).
   B. Tumor size measurements (diameter and calculated volume) throughout course of study and prior to euthanasia (end-point).
   C. Graph of percent tumor diameter change for each condition (compare to Figure 4B and S10C).
   D. Images of H&E stained sections from a random selection of tumors (indicate neovessels and vascular degeneration) (compare to Figure S10B and S10D).
   E. Pathologist's report detailing the evaluation of the stained tumor sections.

## Confirmatory analysis plan

This replication attempt will perform the statistical analyses listed below, compute the effect sizes, compare them against the reported effect size in the original paper and use a meta-analytic approach to combine the original and replication effects, which will be presented as a Forest plot.

1. Statistical analysis of the replication data:
   A. Comparison of percent diameter change in control versus *tk*-expressing tumors.
      i. The diameters of tumors directly before GCV treatment will serve as individual baselines for each mouse. Tumor measurements made 4 weeks after GCV treatment will be subtracted from the baseline measurements for each tumor, and a percent change in diameter will be calculated. The mean percent change in each mouse cohort will be analyzed with a one-way ANOVA.
   a. Following the one-way ANOVA, the following planned pairwise comparisons will be made using the Bonferroni correction to account for multiple comparisons:
   1. For mice implanted with GSC83 neurospheres:
   A. Tie2-*tk* versus PGK-*tk*.

 B. Tie2-*tk* versus empty vector.

 C. PGK-*tk* versus empty vector.

 D. Untransduced (wild-type) versus empty vector (sensitivity).

 b. Following the one-way ANOVA, the following planned pairwise comparisons will be made using the Bonferroni correction to account for multiple comparisons:

 1. For mice implanted with U87MG cells:

 A. Tie2-*tk* versus PGK-*tk*.

 B. Tie2-*tk* versus empty vector (sensitivity).

 C. PGK-*tk* versus empty vector.

 D. Untransduced (wild-type) versus empty vector (sensitivity).

 ii. The authors originally examined the percent changes in tumor diameter between mouse treatment groups using multiple uncorrected two-tailed *t*-tests. We will replicate their *t*-tests, but also use Bonferroni-corrected *t*-tests within the framework of the ANOVA.

B. Comparison of percent volume change in control versus *tk*-expressing tumors.

 i. Differences in percent volume change of tumors before and after GCV treatment will be analyzed as described for percent diameter change above.

C. Comparison of tumor growth rates.

 i. We will measure tumor growth rates across all mouse cohorts over the length of the study, both before and after GCV treatment. These data were not analyzed in the original study, so we consider them exploratory data. We will plot tumor growth curves for each animal and calculate the area under the curve (AUC) before and after GCV treatment. We will perform an ANCOVA on the different treatment groups to evaluate the AUC after GCV treatment, with the baseline (AUC before GCV treatment) included as the covariate. Further, we will perform Bonferroni corrected *t*-tests for pairwise comparisons between controls and *tk*-expressing tumors.

## Known differences from the original study

The methods section in the original paper stated that mice were dual-injected with both control and Tie2-*tk* expressing neurospheres into the right and left flanks, respectively, and bilateral tumors were allowed to form. However, subsequent dialogue with the authors clarified that mice actually only received a single injection, as dual injections often led to problematic differences in tumor growth rates. Therefore, we will be using a single-injection model, where mice will be either injected with *tk* vectors or controls, but not both simultaneously. We will only be comparing GSC83-derived cell lines and U87MG-derived cell lines, excluding the other GSC lines used in the original study. Along with measuring differences in tumor diameter, we will also be measuring tumor volume throughout the course of the study. In the original study, it was not specified how many tumors were harvested and histologically analyzed. We have elected to harvest a random subset of tumors that represent all treatment groups. All known differences in reagents and supplies are listed in the 'Materials and reagents' section above, with the originally used item listed in the 'Comments' section. All differences have the same capabilities as the original and are not expected to alter the experimental design.

## Provisions for quality control

The genetic integrity, mycoplasma-free and rodent pathogen-free purity, and efficient viral transduction of each cell line used in this experiment have been previously validated in Protocols 1 and 2. All mice will be handled and housed in accordance with the Institutional Animal Care and Use Committee (IACUC). All data obtained from the experiment—raw data, data analysis, control data, and quality control data—will be made publicly available, either in the published manuscript or as an open access dataset available on the Open Science Framework (https://osf.io/mpyvx/).

## Power calculations

Unless otherwise stated, all data values are derived from the original paper, or were provided by the original authors.

## Protocol 1

Summary of original data provided by Ricci-Vitiani et al.

| Normalized *Tie2* expression across cell lines (Figure S11C) | Mean | SD | *n* |
|---|---|---|---|
| GSC83 | 0.035362 | 0.012455 | 2 |
| U87MG | 0.018498 | 0.010397 | 2 |
| HMVEC | 1.317634 | 0.021 | 2 |

## Test family

1. ANOVA: fixed effects, omnibus, one-way, with alpha error of 0.05.
   A. ANOVA F-test statistic (performed with GraphPad Prism, version 6.0).
   B. Partial $\eta^2$ calculated from *Lakens (2013)*.

   Power calculations (performed with G*Power software, version 3.1.7 [*Faul et al., 2007*])

| F (Dfn, Dfd) | Partial $\eta^2$ | Effect size f | A priori power | Total sample size |
|---|---|---|---|---|
| F (2, 3) = 4732 | 0.999683 | 56.15669 | 99.9% | 6* (2 per group) |

*A minimum of three samples per group will be used, making the total sample size 9.

## Test family

1. Two-tailed, unpaired *t*-test, with alpha error of 0.05 (Fisher's LSD).
   A. Power calculations (performed with G*Power software, version 3.1.7 [*Faul et al., 2007*])

| Group 1 | Group 2 | Effect size d | A priori power | Group 1 sample size | Group 2 sample size |
|---|---|---|---|---|---|
| GSC83 | HMVEC | 83.68295 | >99.9% | 2* | 2* |
| U87MG | HMVEC | 84.78352 | >99.9% | 2* | 2* |
| GSC83 | U87MG | 3.070892# | 80.0% | 3 | 3 |

*A minimum of three samples per group will be used.
#This is a sensitivity calculation. The original effect size is 1.100569.

# Protocol 2

Not applicable. Power calculations are not necessary for this protocol.

# Protocol 3

Summary of original data provided by Ricci-Vitiani et al.

| Impaired tumor growth of GSC83 xenograft following ganciclovir treatment (Figure 4B) | Mean (% diameter change) | SD | *n* |
|---|---|---|---|
| Wild-type | 7.8 | 5.2 | 3 |
| Empty vector | 8.5 | 5.0 | 3 |
| Tie2-*tk* | −13.2 | 5.9 | 3 |
| PGK-*tk* | −28.7 | 3.4 | 4 |

## Test family

1. ANOVA: fixed effects, omnibus, one-way, with alpha error of 0.05.
   A. ANOVA F-test statistic (performed with GraphPad Prism, version 6.0).
   B. Partial $\eta^2$ calculated from *Lakens (2013)*.
   C. Power calculations (performed with G*Power software, version 3.1.7 [*Faul et al., 2007*])

| F (Dfn, Dfd) | Partial $\eta^2$ | Effect size f | A priori power | Total sample size |
|---|---|---|---|---|
| F (3, 9) = 48.38 | 0.941612 | 4.015856 | 99.9% | 8* (2 per group) |

*A total sample size of 20 will be used based on the planned comparisons.

## Test family

1. Two-tailed, unpaired *t*-test, with alpha error of 0.0125 (Bonferroni's correction).
   A. Power calculations (performed with G*Power software, version 3.1.7 [*Faul et al., 2007*])

| Group 1 | Group 2 | Effect size *d* | A priori power | Group 1 sample size | Group 2 sample size |
|---|---|---|---|---|---|
| PGK-*tk* | Tie2-*tk* | 3.221481 | 93.9% | 5 | 5 |
| Wild-type | Empty vector | 2.640807* | 80.0% | 5 | 5 |
| Empty vector | Tie2-*tk* | 4.510074 | 98.1%# | 4# | 4# |
| Empty vector | PGK-*tk* | 7.731555 | 99.8%§ | 3§ | 3§ |

*This is a sensitivity calculation. The original effect size is 0.1454863.
#Five per group will be used based on the PGK-*tk* versus Tie2-*tk* comparisons making the power 99.9%.
§Five per group will be used based on the PGK-*tk* versus Tie2-*tk* comparisons making the power 99.9%.

## Summary of original data

1. Values estimated from graph in Figure S10C

| Impaired tumor growth of U87MG xenograft following ganciclovir treatment (Figure S10C) | Mean (% diameter change) | SD | *n* |
|---|---|---|---|
| Empty vector | 38.1 | 7.7 | 3 |
| Tie2-*tk* | 36.1 | 11.4 | 3 |
| PGK-*tk* | −49.0 | 7.9 | 3 |

Note: We are including an additional negative control in this experiment (wild-type untransduced cells). We performed these calculations with the assumption that wild-type untransduced cells will have similar values as empty vector control.

## Test family

1. ANOVA: fixed effects, omnibus, one-way, with alpha error of 0.05.
   A. ANOVA F-test statistic partial η² (performed with R software, version 3.1.2 [*R Development Core Team, 2014*]).
   B. Power calculations (performed with G*Power software, version 3.1.7 [*Faul et al., 2007*])

| F (Dfn, Dfd) | Partial η² | Effect size *f* | A priori power | Total sample size |
|---|---|---|---|---|
| F (3, 8) = 72.111 | 0.964339 | 5.200177 | 99.9% | 8* (2 per group) |

*A total sample size of 12 will be used based on the planned comparisons.

## Test family

1. Two-tailed, unpaired *t*-test, with alpha error of 0.0125 (Bonferroni's correction).
   A. Power calculations (performed with G*Power software, version 3.1.7 [*Faul et al., 2007*])

| Group 1 | Group 2 | Effect size *d* | A priori power | Group 1 sample size | Group 2 sample size |
|---|---|---|---|---|---|
| PGK-*tk* | Tie2-*tk* | 9.651957 | 99.9% | 3 | 3 |
| Wild-type | Empty vector | 4.521980 | 80.0%* | 3 | 3 |
| Empty vector | Tie2-*tk* | 4.521980 | 80.0%# | 3 | 3 |
| Empty vector | PGK-*tk* | 9.878795 | 99.9% | 3 | 3 |

*This is a sensitivity calculation. There is no original effect size.
#This is a sensitivity calculation. The original effect size is 0.226838.

## Acknowledgements

The Reproducibility Project: Cancer Biology core team would like to thank the original authors, including Ruggero De Maria, Roberto Pallini, and most especially, Lucia Ricci-Vitiani, for generously sharing critical information as well as reagents to ensure the fidelity and quality of this replication attempt. We thank Courtney Soderberg at the Center for Open Science for assistance with statistical analyses. We would also like to thank the following companies for generously donating reagents to the Reproducibility Project: Cancer Biology: American Type Culture Collection (ATCC), BioLegend, Cell Signaling Technology (CST), Charles River Laboratories, Corning Incorporated, DDC Medical, EMD Millipore, Harlan Laboratories, LI-COR Biosciences, Mirus Bio, Novus Biologicals, System Biosciences, and Sigma–Aldrich.

## Additional information

### Group author details

**Reproducibility Project: Cancer Biology**

Elizabeth Iorns: Science Exchange, Palo Alto, California; William Gunn: Mendeley, London, United Kingdom; Fraser Tan: Science Exchange, Palo Alto, California; Joelle Lomax: Science Exchange, Palo Alto, California; Timothy Errington: Center for Open Science, Charlottesville, Virginia

### Competing interests

DC: Noble Life Sciences is a Science Exchange-associated lab. DS: BioFactura is a Science Exchange-associated lab. RP:CB: We disclose that Elizabeth Iorns, Fraser Tan, and Joelle Lomax are employed by and hold shares in Science Exchange Inc. The experiments presented in this manuscript will be conducted by DC at Noble Life Sciences and DS at BioFactura, which are both Science Exchange-associated laboratories. The other authors declare that no competing interests exist.

### Funding

| Funder | Author |
| --- | --- |
| Laura and John Arnold Foundation | Reproducibility Project: Cancer Biology |

The funder had no role in study design, data collection and interpretation, or the decision to submit the work for publication.

### Author contributions

DC, DS, Conception and design; NM, Drafting or revising the article; RP:CB, Conception and design, Drafting or revising the article

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
