## [Decision Letter]

Thank you for sending your work entitled “Registered report: Tumour
vascularization via endothelial differentiation of glioblastoma stem-like cells”
for consideration at *eLife*. Your article has been favorably evaluated
by Sean Morrison (Senior editor), a Reviewing editor, four reviewers, and a
biostatistician. Two of the reviewers, Yoshiaki Kubota and Ruggero De Maria, have agreed
to share their identity.

The Reviewing editor and the reviewers discussed their comments before we reached this
decision, and the Reviewing editor has assembled the following comments to help you
prepare a revised submission.

As part of The Reproducibility Project: Cancer Biology, Chroscinski et al. propose to
replicate key experiments from “Tumour vascularization via endothelial
differentiation of glioblastoma stem-like cells” by Ricci-Vitiani and colleagues,
published in Nature in 2010. The original Nature manuscript demonstrated that
endothelial cells bearing the same genomic alteration as cancer cells may show a
different sensitivity to conventional anti-angiogenic treatments and suggests the
possibility of targeting the process of GSC differentiation into endothelial cells as a
new therapeutic option in cancer treatment.

The authors accurately summarize the Nature paper and the proposed experiment using GSCs
transduced with a Tie2-tk construct sufficiently replicates the key finding in that
Nature paper.

The following questions should be addressed in a resubmission:

1) A directly conflicting paper by Cheng et al. (Cell, 2013) discussed that the
discrepancy is attributed to possible activation of Tie2 promoter in pericytes, which
may eliminate pericytes as well. Do the authors have any discussion on this issue?

2) Within protocol 1, the authors plan to use HMVEC instead of HUVEC as previously
outlined in the original study. Endothelial cells of different lineage vary greatly in
their gene expression profile in particular between HUVEC and HMVEC. As HUVECs are
commercially available, the authors should provide a clearer explanation in their change
of reagents.

3) To maximize the quality of the replication, we would suggest that the authors
establish more precise criteria for neurosphere passage. Protocol 1.1.a.ii states that
“GSC83 cells should be subcultured when neurosphere size prevents the needed
provision of nutrients”. This criteria is subjective, and viability,
differentiation potential, infection efficiency etc of glioma stem cells can all be
influenced by the abundance or lack of nutrients. Always plating a fixed cell number and
subculturing after a fixed period of time could be one way to avoid variability in
culture methods.

Statistical comments:

Overall, I have no major concerns over the statistics.

1) For protocols 1 and 3, the aim is to replicate the detection of the very large
effects seen in the original report. The original report used very small numbers so both
the reported effect size and the SD will have very poor precision. With a sample size of
2-4 per group it is impossible to judge whether the data points have been sampled from a
normal distribution so the p-value strongly relies on the assumption of normality (which
is not so important for large sample sizes). However, the researchers presumably have
extensive prior experience of using such techniques in similar situations and hence
expect to see very large effects with no overlap of the distributions of values observed
in each group. So I judge that the very small sample sizes, provided no data are
missing, will be sufficient to answer these questions.

2) It seems reasonable to use the Bonferroni adjusted significance levels as the same
sample is used for several comparisons.

3) When it comes to presenting the results and comparing with the original report, a
Forest plot is appropriate but this should not include an overall summary statistic
(combining the original and replicate).

4) The method used to compare the growth rates seems rather convoluted. Why not put the
repeated measurements into a full linear model with the baseline included as a
co-variate?

---

## [Author Response]

*1) A directly conflicting paper by Cheng et al. (Cell, 2013) discussed that the
discrepancy is attributed to possible activation of Tie2 promoter in pericytes, which
may eliminate pericytes as well. Do the authors have any discussion on this
issue*?

As reviewed in the Introduction, many conflicting reports exist within the field of
tumor-derived vasculature in GBM. We believe the data presented in Cheng et al. provide
a very interesting alternative hypothesis that may help to explain the results achieved
by Ricci-Vitiani and colleagues. We have expanded our discussion of this paper in the
Introduction, to better outline the results of Cheng et al. We will be sure to highlight
these data again in the Replication Study as a possible alternative mechanism to
understand the results of the replication.

*2) Within protocol 1, the authors plan to use HMVEC instead of HUVEC as
previously outlined in the original study. Endothelial cells of different lineage
vary greatly in their gene expression profile in particular between HUVEC and HMVEC.
As HUVECs are commercially available the authors should provide a clearer explanation
in their change of reagents*.

As demonstrated in Supplementary Figure 11C (13), Ricci-Vitiani and colleagues measured Tie2
expression in both HUVEC and HMVEC cell lines. Both cell lines served as a positive
control, demonstrating high levels of Tie2 expression. According to our correspondence
with Ricci-Vitiani and colleagues, they included HMVEC cells as an additional control at
the behest of their own reviewers for the original Nature manuscript. These reviewers
felt that including an adult endothelial cell line constituted a more accurate control
for adult-derived glioblastoma neurospheres than did the fetal-derived HUVEC cells.
Thus, we have opted to also include this more-relevant control cell line. We are
obtaining the same adult HMVEC cell line as used by the original authors to enable a
direct comparison of gene expression.

*3) To maximize the quality of the replication, we would suggest that the authors
establish more precise criteria for neurosphere passage. Protocol 1.1.a.ii states
that “GSC83 cells should be subcultured when neurosphere size prevents the
needed provision of nutrients”. This criteria is subjective, and viability,
differentiation potential, infection efficiency etc of glioma stem cells can all be
influenced by the abundance or lack of nutrients. Always plating a fixed cell number
and subculturing after a fixed period of time could be one way to avoid variability
in culture methods*.

We thank the reviewers for these suggestions. Upon further correspondence with
Ricci-Vitiani and colleagues, we have updated the Registered Report to include more
details pertaining to the culture of GSC#83, including defined cell plating
numbers and subculturing schedule. We will be obtaining the same culture that was
initially derived by the authors, and using the same conditions they use for this
culture.

*Statistical comments*:

[…]

*3) When it comes to presenting the results and comparing with the original
report, a forest plot is appropriate but this should not include an overall summary
statistic (combining the original and replicate)*.

We disagree with the reviewer and as described in Valentine et al., 2011 and Bumming,
2012, combining multiple studies using a meta-analytic approach is a statistical option
that can be employed to describe all of the available evidence about a given effect
size. Specifically we will utilize a random effects meta-analysis because this approach
assumes that the effects vary due to known and unknown characteristics of the
studies.

*4 ) The method used to compare the growth rates seems rather convoluted. Why not
put the repeated measurements into a full linear model with the baseline included as
a co-variate*?

We thank the reviewer for this helpful suggestion. We have changed the method used to
compare the growth rates of the different cohorts. We will perform an ANCOVA to analyze
the differences in the tumor growth curves after GCV treatment using the tumor growth
curves before GCV treatment (baseline) as the covariate. We will also use the area under
the curve to quantify and compare the tumor growth curves.